# *Babesia bovis* Rad51 ortholog influences switching of *ves* genes but is not essential for segmental gene conversion in antigenic variation

Erin A. Mack [1☯¤], Massimiliano S. Tagliamonte [2,3☯], Yu-Ping Xiao[1], Samantha Quesada[1], David R. Allred [1,2,3,4]*

1 Department of Infectious Diseases and Immunology, University of Florida, Gainesville, Florida, United States of America, 2 Department of Pathology, Immunology, and Laboratory Medicine, University of Florida, Gainesville, Florida, United States of America, 3 Emerging Pathogens Institute, University of Florida, Gainesville, Florida, United States of America, 4 Genetics Institute, University of Florida, Gainesville, Florida, United States of America

☯ These authors contributed equally to this work.
¤ Current address: Department of Biological Sciences, University of Idaho, Moscow, Idaho, United States of America
* allredd@ufl.edu

**Data Availability Statement:** All relevant data are within the manuscript and supporting information files, or is in the form of raw sequence data

## Abstract

The tick-borne apicomplexan parasite, *Babesia bovis*, a highly persistent bovine pathogen, expresses VESA1 proteins on the infected erythrocyte surface to mediate cytoadhesion. The cytoadhesion ligand, VESA1, which protects the parasite from splenic passage, is itself protected from a host immune response by rapid antigenic variation. *B. bovis* relies upon segmental gene conversion (SGC) as a major mechanism to vary VESA1 structure. Gene conversion has been considered a form of homologous recombination (HR), a process for which Rad51 proteins are considered pivotal components. This could make BbRad51 a choice target for development of inhibitors that both interfere with parasite genome integrity and disrupt HR-dependent antigenic variation. Previously, we knocked out the Bb*rad51* gene from the *B. bovis* haploid genome, resulting in a phenotype of sensitivity to methyl-methane sulfonate (MMS) and apparent loss of HR-dependent integration of exogenous DNA. In a further characterization of BbRad51, we demonstrate here that ΔBb*rad51* parasites are not more sensitive than wild-type to DNA damage induced by γ-irradiation, and repair their genome with similar kinetics. To assess the need for BbRad51 in SGC, RT-PCR was used to observe alterations to a highly variant region of *ves1*α transcripts over time. Mapping of these amplicons to the genome revealed a significant reduction of in situ transcriptional switching (isTS) among *ves* loci, but not cessation. By combining existing pipelines for analysis of the amplicons, we demonstrate that SGC continues unabated in ΔBb*rad51* parasites, albeit at an overall reduced rate, and a reduction in SGC tract lengths was observed. By contrast, no differences were observed in the lengths of homologous sequences at which recombination occurred. These results indicate that, whereas BbRad51 is not essential to babesial antigenic variation, it influences epigenetic control of *ves* loci, and its absence significantly reduces successful variation. These results necessitate a

available as NCBI BioProject #PRJNA357248 (accession numbers SRR5110992-SRR5111003).

**Funding:** This work was supported by funds from National Institutes of Health grants R01 AI0055864 (DRA) and T32 AI0007110 (EAM), USDA Animal Health project FLA-VME-005011 (DRA), and U.F. College of Veterinary Medicine (DRA). The funders had no role in study design, data collection and analysis, decision to publish, or preparation of the manuscript.

**Competing interests:** The authors have declared that no competing interests exist.

reconsideration of the likely enzymatic mechanism(s) underlying SGC and suggest the existence of additional targets for development of small molecule inhibitors.

## Author summary

*B. bovis* establishes highly persistent infections in cattle, in part by using cytoadhesion to avoid passage through the spleen. While protective, a host antibody response targeting the cytoadhesion ligand is quickly rendered ineffective by antigenic variation. In *B. bovis*, antigenic variation relies heavily upon segmental gene conversion (SGC), presumed to be a form of homologous recombination (HR), to generate variants. As Rad51 is generally considered essential to HR, we investigated its contribution to SGC. While diminishing the parasite's capacity for HR-dependent integration of exogenous DNA, the loss of BbRad51 did not affect the parasite's sensitivity to ionizing radiation, overall genome stability, or competence for SGC. Instead, loss of BbRad51 diminished the extent of in situ transcriptional switching (isTS) among *ves* gene loci, the accumulation of SGC recombinants, and the mean lengths of SGC sequence tracts. Given the overall reductions in VESA1 variability, compromise of the parasite's capacity for in vivo persistence is predicted.

## Introduction

The apicomplexan parasite, *Babesia bovis*, is a quick-change artist with the ability to rapidly alter proteins it expresses on the infected erythrocyte surface. This ability is needed because, during asexual reproduction *B. bovis* cytoadheres to the capillary and post-capillary venous endothelium within its bovine mammalian host [1–3]. It is thought that this behavior allows this microaerophilic parasite to avoid splenic clearance and to complete asexual development under hypoxic conditions, analogous to the human malarial parasite, *Plasmodium falciparum* [4]. Cytoadhesion is mediated by heterodimeric variant erythrocyte surface antigen-1 (VESA1) proteins, which are exported by the parasites to the erythrocyte where they integrate into the erythrocyte membrane [5, 6]. Cytoadhesion is compromised by a host antibody response targeting VESA1, preventing and potentially reversing cytoadhesion [6]. However, *B. bovis* has evolved the ability to rapidly vary the structure and antigenicity of VESA1 polypeptides, abrogating recognition by existing antibodies [7–10]. Antigenic variation in *B. bovis* involves the *ves* multigene family encoding VESA1a and 1b polypeptides [9, 11, 12], and possibly the *smorf* multigene family [13, 14]. *B. bovis* intraerythrocytic stages reproduce asexually, with a haploid genome of only 8–8.5 Mbp [13]. Despite its small genome size, approximately 135 genes comprise the *B. bovis ves* multigene family, amounting to approximately 4.7% of all coding sequences [10, 13]. Transcription of *ves* genes is monoparalogous, arising from a single *ves* locus at any one time (but typically involves transcription of both a *ves*1α and *ves*1β gene from the same locus to encode both subunits), whereas the remainder of the family remains transcriptionally inactive [15]. In situ transcriptional switching (isTS) from one *ves* locus to another over time has been implicated in *B. bovis* antigenic variation [10, 15, 16], although segmental gene conversion (SGC) is the only mechanism of variation critically demonstrated in this parasite to date [12]. Progressive replacement of short sequence patches within the actively transcribed *ves* genes by SGC, yields *ves* genes (and VESA1 polypeptides) that are mosaics comprised of sequences from many *ves* loci [11, 12]. The short lengths of the SGC conversion

tracts, ability to acquire sequences from any chromosome, and involvement of two similarly variant subunits may enable this gene family to provide practically unlimited diversity in epitope structure [5, 7, 9, 11, 12].

Canonical gene conversion is a form of homologous recombination (HR)-mediated DNA repair. In this process a damaged sequence is repaired by incorporating duplicated homologous sequences from an undamaged allele or paralog to replace the damaged sequences. At least three models have been proposed to explain this process (reviewed in [17]). Common to all models is the assembly of a repair complex at the site of damage. Activities of the repair complex include 5' to 3' resection of the broken ends of the damaged molecule, and stabilization of the single-stranded 3' ends by assembly of RPA ssDNA-binding proteins onto the strands (reviewed in [17, 18]). Rad52 may stabilize the RPA and maintain spatial proximity of the broken ends [19]. The Rad52-RPA complexes then are replaced by Rad51 protein, which forms helical filaments on the ssDNA strands. The Rad51-ssDNA complexes, together with Rad54, mediate both a search for homologous sequences elsewhere in the genome and strand invasion when such sequences are found [20–22]. Once found, single-stranded invasion of the identified sequence allows sequence acquisition by extension of the 3' end of the invading strand. Depending upon the model, the inter-chromosomal entanglement involves either one or two Holliday junction structures that are resolved to yield two independent chromosomes again. Sequence acquisition may occur with or without crossover, depending upon how the junction structures are resolved [23]. Like canonical gene conversion, we hypothesized SGC to occur via HR, a possibility consistent with the stretches of homologous sequence flanking SGC tracts that are shared between donor and recipient. However, other factors call this into question. For example, the reasons for consistently short conversion tracts are unknown, and crossover appears to be a rare outcome in *B. bovis* [10]. These traits suggest that minimal end resection, acquisition of only short tracts of differing sequence, and rapid resolution of intermolecular junction structures all may define this process mechanistically. Among the existing models of HR, those most consistent with these traits are synthesis-dependent strand annealing, either with rapid disentanglement of the invading strand(s), or double-strand break repair but with convergent branch migration and junction dissolution. By contrast, resolution of distal Holliday junctions would be expected to result in frequent crossover events [17]. Alternatively, a more exotic explanation may hold, such as the involvement of a template-switching repair polymerase, but no direct evidence yet supports this possibility.

Rad51 is considered essential to HR and the gene conversion process in eukaryotes. Organisms with defective Rad51 consistently suffer reduced viability and enhanced sensitivity to environmental insult [24–29]. For example, knockout of the *Rad51* gene in mice resulted in embryonic lethality [30], whereas loss of the Tb*rad51* gene in the kinetoplastid parasite, *Trypanosoma brucei*, yielded parasites that were compromised in growth and hypersensitive to methyl methanesulfonate (MMS) [28]. Interestingly, Tb*rad51* knockout parasites continued to undergo variation of their variant surface glycoprotein (*VSG*) genes, both by isTS and gene conversion mechanisms, but the rate at which variation occurred was slowed dramatically. This observation suggested a role for TbRad51 in facilitating or regulating trypanosomal antigenic variation, but not an essential role in catalysis. Moreover, TbRad51-independent mechanisms may act in trypanosomal antigenic variation. Recently, we knocked out the Bb*rad51* gene of *B. bovis*. Unlike higher eukaryotes, there was no apparent effect on parasite viability or growth. However, parasites were made hypersensitive to MMS and failed to integrate exogenous DNA, suggesting defects in HR [31]. Given the importance of SGC to *B. bovis* antigenic variation and survival, and of Rad51 to gene conversion and HR, we investigated further the interplay between DNA repair and antigenic variation. We provide evidence that overall DNA repair remains highly robust in the absence of BbRad51. SGC also continues, albeit at an

overall reduced rate, concomitant with a significant reduction in isTS. We hypothesize that these results reflect unique roles for BbRad51 in antigenic variation, and suggest that alternative enzymes catalyze recombination during SGC.

## Results

### Bbrad51 knockout parasites behave similarly to wild-type in response to DNA damage from γ-irradiation

Previously, we knocked out the Bb*rad51* gene, resulting in a parasite phenotype of MMS-sensitivity and apparent loss of the ability to integrate exogenous DNA by HR, but not significant difference in rates of growth [31]. A second measure of phenotype commonly used in studies of DNA repair is sensitivity to γ-irradiation. Sensitivity of the knockout parasites to DNA damage was assessed by exposure to γ-irradiation provided by a $Cs^{137}$ source. Wild type CE11 parasites first were titrated for sensitivity, over a range from 10–1230 gray (Gy). A dosage of 100 Gy resulted in approximately 80–90% lethality by 24 hours, but allowed some parasite survival at 48 hours and beyond, whereas at 200 Gy no parasites were observed to survive 48 hours post-irradiation (S1 Fig). Radiation sensitivities of three independently-derived Bb*rad51* knockout lines (all on a CE11 genetic background) therefore were compared with CE11 wild-type parasites over a range from 0–200 Gy. There were no reproducible differences among any of the three knockout lines and CE11 wild type parasites in growth assays (Fig 1).

Because no differential survival phenotype was apparent following DNA damage by γ-irradiation, we asked whether there were any detectable differences in overall rates of DNA repair. To assess this, pulsed-field gel electrophoresis (PFGE) was used to monitor the disintegration of chromosomes and their subsequent reassembly. Parasites were subjected to 100 Gy γ-irradiation, allowed to recover for up to 24 hours, and then were processed for PFGE analysis. In growth experiments, 100 Gy had resulted in killing of 80–90% of the parasites. Consistent with this level of killing, 100 Gy γ-irradiation severely damaged *B. bovis* chromosomes, virtually eliminating full-length chromosomes 3 and 4, and greatly diminishing the proportion of intact chromosomes 1 and 2. Remarkably, by 24 hours post-irradiation both wild-type and knockout parasites had reassembled their genomes into full-length chromosomes of apparently normal size, recovering approximately 50% of non-irradiated control values (Fig 2). Given the 10–20% viability of parasites receiving this dosage (S1 Fig) it is likely that a large proportion is not viable in the longer-term, but remains metabolically active long enough to reassemble chromosomes. These data, when considered together, demonstrate that the loss of BbRad51 has little, if any, effect on the extensive DNA repair required to recover from such damage and suggest that BbRad51-dependent HR plays little role in this type of repair.

### Bbrad51 knockout resulted in apparent reduction of in situ transcriptional switching

Although overall DNA repair following damage from ionizing radiation was not measurably impaired by the loss of BbRad51, sensitivity to MMS and loss of ability to integrate selectable plasmids via long sequence tracts suggested that BbRad51 plays a role in aspects of HR [31], and perhaps in repair of stalled replication forks [32]. Previously, it was demonstrated that SGC is a major mechanism of antigenic variation in *B. bovis* [12]. Given the seemingly conflicting outcomes obtained with MMS and ionizing radiation, we wished to determine whether BbRad51 plays any role in SGC. In order to assess the nature of any changes occurring in transcribed *ves1α* genes of wild type and Bb*rad51* knockout parasites, we adapted a previously published assay in which the highly variant cysteine-lysine-rich domain (CKRD) region of

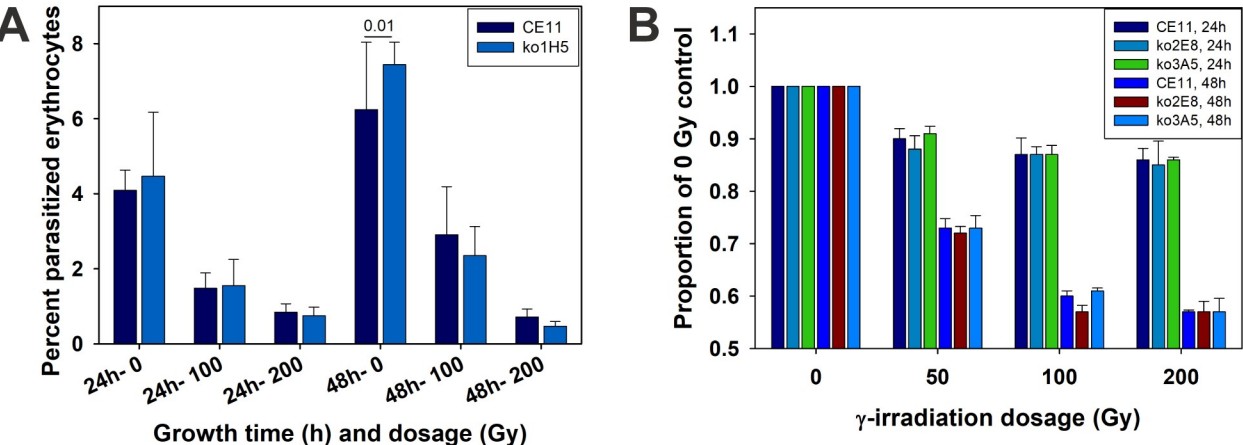

**Fig 1. Survival and growth of Bb*rad51* knockouts following γ-irradiation is not different from wild-type. A**. Parasites were exposed to varying dosages of γ-irradiation, placed back into culture, sampled at 24 and 48h post-irradiation, and percent parasitized erythrocytes determined microscopically. In this individual experiment a significant difference was observed at 48h in the non-irradiated control samples only (p = 0.01). This was not a reproducible difference. This assay shows that there is no increase in percent parasitized erythrocytes over time after 200 Gy exposure. **B**. Comparison of CE11 wild-type with knockout clonal lines ko2[E8] and ko3[A5], determined by a Sybr Green DNA-detection assay [31]. Sybr Green data are plotted as the proportion of the signal exhibited by the non-irradiated control samples (± 1 s.d.). No reproducible significant differences were observed among the parasite lines at these dosages. Note that the ko1[H5] line was derived approximately one year prior to ko2[E8] and ko3[A5] lines and was assessed differently, by microscopic observation of percent parasitized erythrocytes. In each case, this experiment was repeated three times with four replicates per sample. No reproducible differences were observed between *B. bovis* CE11 wild-type and any of the knockout lines.

*ves1*α transcripts is amplified by RT-PCR (Fig 3), and the amplicons undergo deep sequencing [15]. Three immediate subclones of CE11 wild-type and three independent knockout clonal lines were studied; their origin is described in [31]. Total RNAs were collected from each clonal line at one month and five months post-cloning. These two timepoints were used to

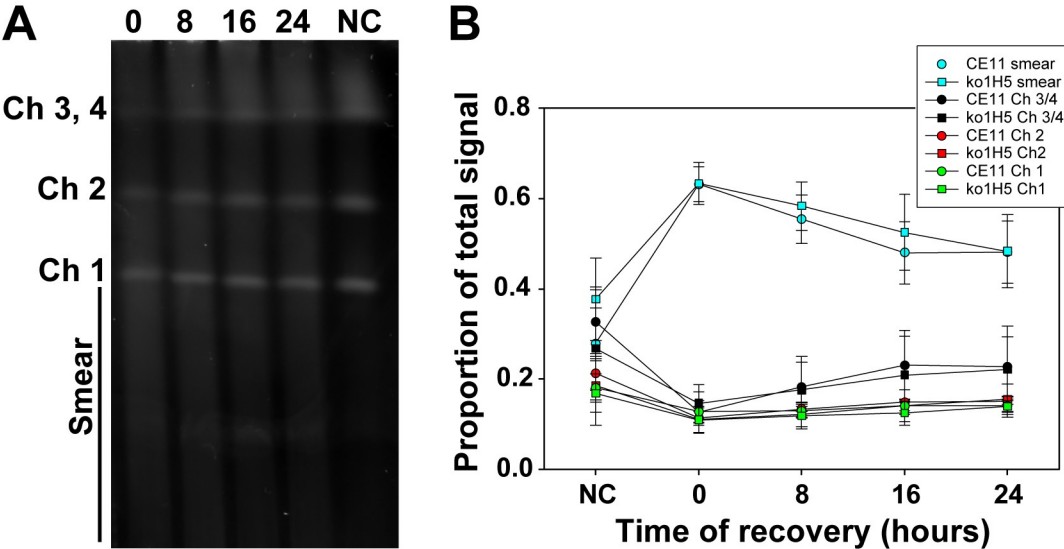

**Fig 2. Chromosomes of γ-irradiated parasites reassembled with the same kinetics.** *B. bovis* CE11 wild type and ko1[H5] clonal line were exposed to 100 Gy γ-irradiation, then allowed allowed to recover for 0, 8, 16, or 24h. **A**. Reassembly of chromosomes was assessed by PFGE (CE11 wild type is shown as an example). **B**. Plot of relative signal intensities of *B. bovis* CE11 wild type and ko1[H5] DNAs within individual chromosomes, and in the "smear" below chromosome 1. The plot includes the means (± 1 s. d.) of four independent experiments. Statistical analyses using Student's T-test revealed no significant differences between wild type and knockout parasites at any time point. NC, non-irradiated control.

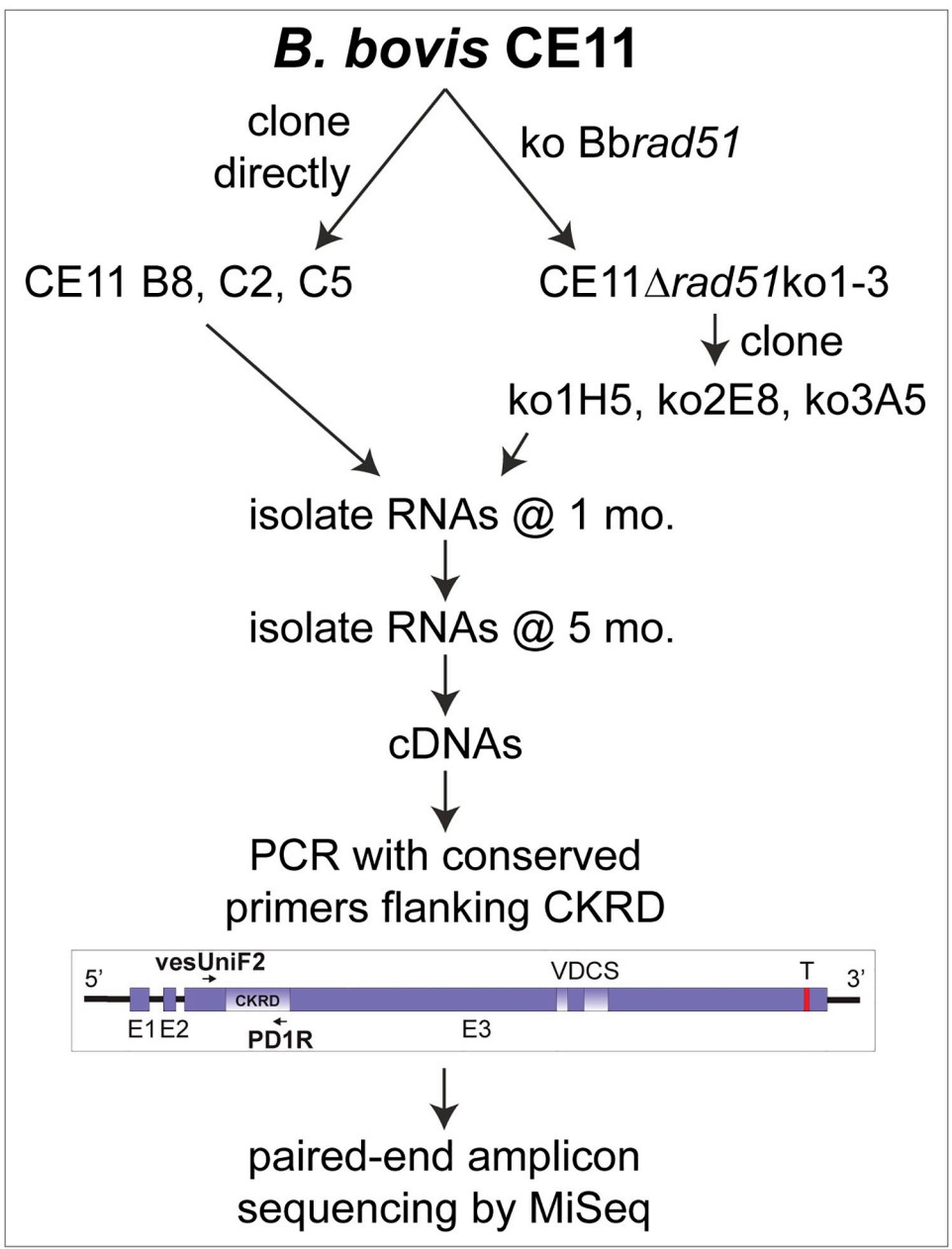

**Fig 3. Flow of the assay for SGC characteristics.** For this experiment, *B. bovis* CE11 was directly subcloned to provide clonal lines CE11[B8], CE11[C2], and CE11[C5]. Three independent Bb*rad51* knockout lines were created and cloned, giving rise to clonal lines ko1[H5], ko2[E8], and ko3[A5]. Total RNAs were collected at 1 month and 5 months post-cloning, and used for cDNA generation. The primer pair *ves*UniF2 and PD1R flanking the highly variant CKRD domain were used to PCR-amplify transcribed *ves1α* transcripts. Amplicons were used to generate paired-end libraries for sequencing by Illumina MiSeq. Unique recombinant sequences were identified bioinformatically. Details for all aspects are found in Methods.

observe for increases in the numbers of unique recombinants over time, and for some analyses were pooled to minimize the loss of unique variants from the population over time. The forward primer, *ves*UniF2, was selected because it represents a sequence almost universally conserved among *ves*1α genes, and in combination with the highly-conserved primer PD1R was anticipated to generate amplicons of approximately 340–460 bp. The flow of this experiment is

shown schematically in Fig 3. By constructing bar-coded paired-end libraries from the amplicons and generating 250 bp reads, reads could be merged with high-confidence overlaps of 55–185 bp. Merged, full-length sequences were obtained from 69.8–84.5% of amplicon reads. Following the removal of ambiguous and low-quality reads, adaptor and primer sequences, and sequences found likely to be PCR chimeras, individual libraries ranged from a minimum of 853,688 to a maximum of 1,738,649 merged reads. Mean merged read lengths ranged from $326.2 \pm 60.3$ to $334.7 \pm 52.9$ bp. To determine their probable loci of origin, reads were mapped (using non-global settings) onto the *B. bovis* C9.1 line genomic sequence (available at Wellcome Trust Sanger Institute; ftp://ftp.sanger.ac.uk/pub/pathogens/Babesia/). The C9.1 line genome was used because we do not currently have a high quality genome for the CE11 line. However, as these are closely-related clonal sibling lines [12] this allowed us to identify the probable locus of origin for nearly all reads. In all six knockout and wild-type lines, the earlier time-point *ves1α* transcripts mapped predominantly, sometimes almost solely, to a single locus (Fig 4A), consistent with prior observations of monoparalogous *ves* gene transcription in the C9.1 clonal line [15]. At the latter time-point, lines ko1[H5] and CE11[B8] continued to transcribe almost solely from the original locus, and all lines still transcribed most heavily from the original locus. Minor but significant subpopulations were detected in all lines that had switched to transcription from alternative loci. Without immune pressure there is no obvious selection for parasites expressing specific VESA1a isoforms to predominate. Regardless, detectable transcription occurred from more alternative loci in wild type CE11 subclones B8, C2, and C5 (ranging from $28.7 \pm 4.0$ at 1 month to $33.3 \pm 1.5$ loci at 5 months) than were observed for the three knockout lines, which ranged from $22.o \pm 3.5$ at 1 month to $23.3 \pm 1.5$ at 5 months (Fig 4B). While transcribing most heavily from a single locus, the CE11 C5 subclone (as a population) also transcribed significantly from several alternative loci at the early time-point, but by the 5-month time-point transcription levels had been reduced from all but the single, major locus. Interestingly, the same alternative loci seemed to dominate as sites to which switching occurred (S2 Fig), suggesting a hierarchy in locus transcription. However, nothing can be inferred from these data regarding an order in switching like that documented for *P. falciparum var* genes [33].

## Bbrad51 knockout reduces overall variation without abrogating segmental gene conversion

Accurate identification and characterization of SGC tracts would be most readily achieved by directly mapping amplicon sequences against a reference genome. Although the CE11 and C9.1 lines are closely related sibling clonal progeny of the MO7 clonal line, they have quite different histories [12]. Because of the nature and rapidity of SGC many loci would have been extensively modified. The ideal would be to map to a high quality genome from each subclone, so that unique variants in each line would be known, but this approach has its own limitations. Given that such a genome relies upon consistent sequence this would require cloning of the parasite, effectively selecting one variant from a pool of possible variants. This would bias the outcome toward apparently greater or lesser similarity depending upon how representative the cloned parasite is of the major population. One potential approach to solve this difficulty might be to use conserved *ves* family sequence patches [12] to perform a targeted sequence enrichment prior to library construction, followed by deep sequencing [34]. The alternative we chose was to observe for recombination among loci represented by the *ves1α* transcript amplicons, considering only sequences unique to an individual line, on a line by line basis. By taking this approach, we could assess for recombination among sequences that were definitively present in each line at the time the experiment was performed. Using an RT-PCR

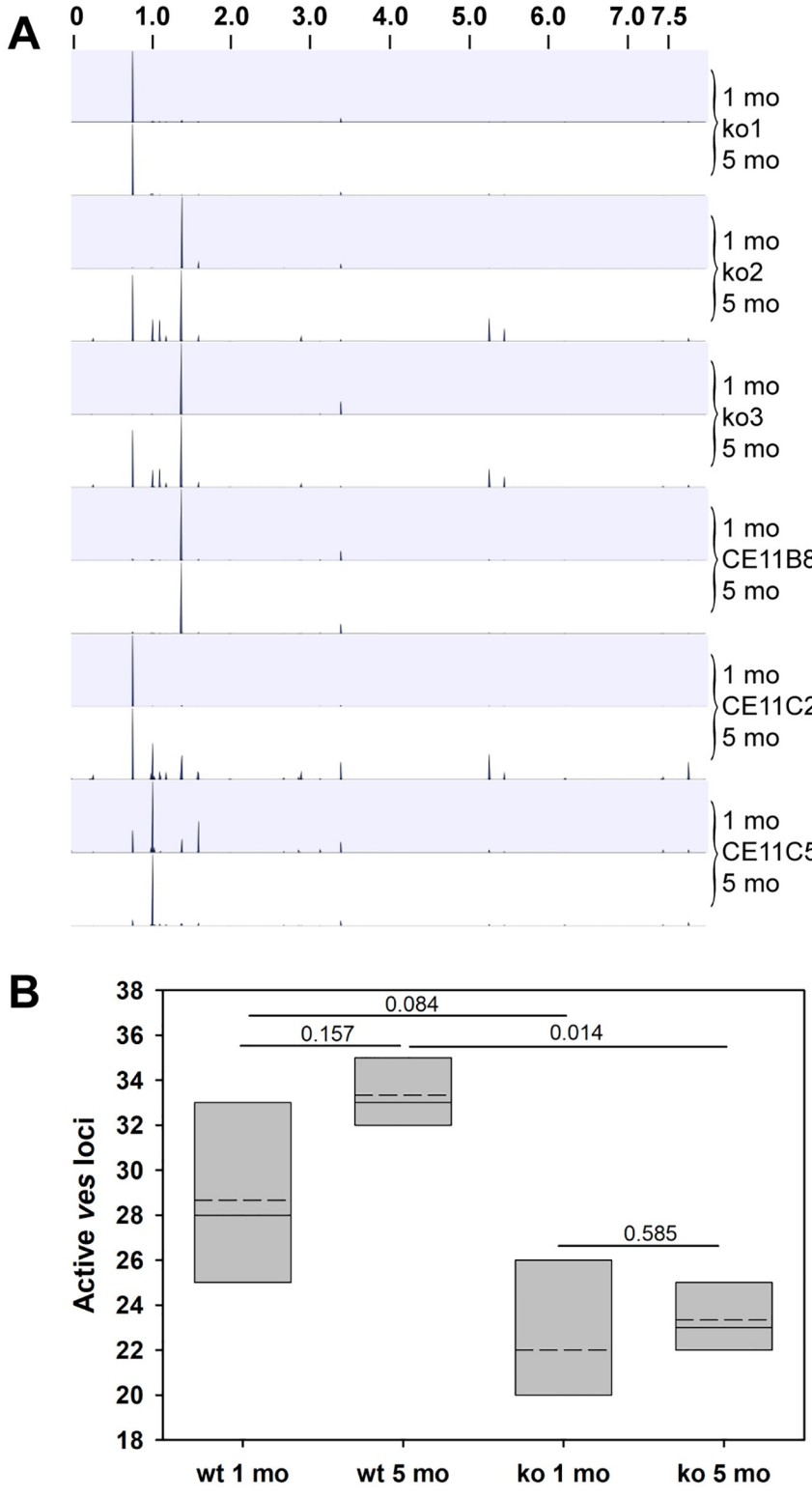

**Fig 4. Variation arose over time in the sequences of *ves*1α transcripts.** RT-PCR amplification of the highly variable CKRD domain of *ves*1α transcripts and deep sequencing of amplicons was used to assess variation arising over time. Three freshly cloned lines of *B. bovis* CE11 wild type (lines CE11[B8], CE11[C2], and CE11[C5]), and one clonal line from each of three independent CE11Δ*rad51* knockout lines were observed 1 and 5 months post-cloning. **A**. All merged amplicon sequences were mapped to the *B. bovis* C9.1 genome, using CLC Genomics Workbench. Mapping tracks are

shown for each sample. The X axis represents the 8 Mbp *B. bovis* C9.1 genome as a single linear sequence, the Y axis represents the proportion of sequences aligning to a given *ves* locus within the genome, and the alternating blue and white bands represent data from 1 month (blue) and 5 month (white) time-points for each parasite line. The area of each peak is proportional to the numbers of reads mapping to that locus relative to the total. Transcripts from many activated loci are present in too low abundance to be observable in this plot. These data confirm earlier results suggesting that *ves* transcription is monoparalogous [15], but also show that isTS appears to occur over time. **B**. Numbers of transcriptionally active *ves* loci detected at each time point is shown (data is pooled for knockouts and for CE11 samples). Slotted crossbars indicate means, solid crossbars indicate medians, and box boundaries represent 25% and 75% confidence intervals. Statistical significance of differences among samples, based upon one-way ANOVA, are indicated.

strategy employed previously to characterize variation in *ves1*α transcripts [15], we identified all unique transcribed sequences by cluster analysis, then identified representative reads for each cluster. Among those, we then identified sequences for which there was very strong statistical support for them being the result of a true recombination event between two other unique sequences, based upon a consensus of 4 out of 7 statistical analyses (see Methods for details). For comparative analysis, we included only those recombinant sequences found in a single clonal line, on the assumption that any sequences found in more than one line was present prior to the act of parasite cloning and not a result of post-cloning recombination. Information on all identified SGC tracts are provided in S1_File.zip. We propose that most of the unique sequences were in fact recombinant, as a plot of all unique sequences against all statistically-supported recombinant sequences (both normalized per million reads) a regression of $R^2 = 0.9228$ was obtained. Thus, the use of a statistical consensus approach was highly conservative and likely underestimates the true number of recombinant sequences (S3 Fig), but allows for rigorous comparative results.

Variants with strong statistical support as true recombinant SGC tracts were observed among the transcripts of all six clonal lines, but clear distinctions are seen between knockout and wild type. The mean lengths of conversion tracts differed between groups, decreasing from means of 109.01 ± 39.39 in wild type parasites to 91.77 ± 40.78 in knockouts ($p < 0.001$; Fig 5A). The distributions of SGC tract lengths, plotted as cumulative proportions of all SGC tracts from that population, resulted in wild type and knockout medians of 107.0 and 81.0 (S4 Fig; $p < 0.001$). Interestingly, the difference between the two populations arose primarily at tract lengths <150 bp. The two populations were not distinct above that length, suggesting the possibility that more than one mechanism gave rise to the SGC tracts. The number of unique SGC tracts arising per active *ves* locus was not significantly different, ranging from 0.35–1.20 SGC tracts per locus (S5 Fig; $p = 0.22$ among all group comparisons). Although there was a rise in the frequency of recombinants per locus in CE11 wt parasites over time this was not statistically significant ($p = 0.18$). However, with the differences in the numbers of transcriptionally active *ves* loci in wild type and knockout parasites, this led to significantly larger total numbers of unique SGC tracts per million reads among members of the wild type population (Fig 5B; at 5 months, $p < 0.02$). Thus, while the frequency of SGC alterations that may be observed at any given transcriptionally active *ves* locus is approximately constant, the numbers of *ves* loci that are activated is significantly higher in the presence of BbRad51. The number of unique SGC tracts per million reads may be considered a surrogate measure of overall levels of variability in antigen structure presented by the population of parasites. Taken together, these data demonstrate clearly that neither SGC nor apparent isTS to alternative loci is abrogated by knockout of the Bb*rad51* gene, although statistically significant quantitative effects on the lengths of SGC tracts and on the frequency and extent of *ves* locus switching were observed.

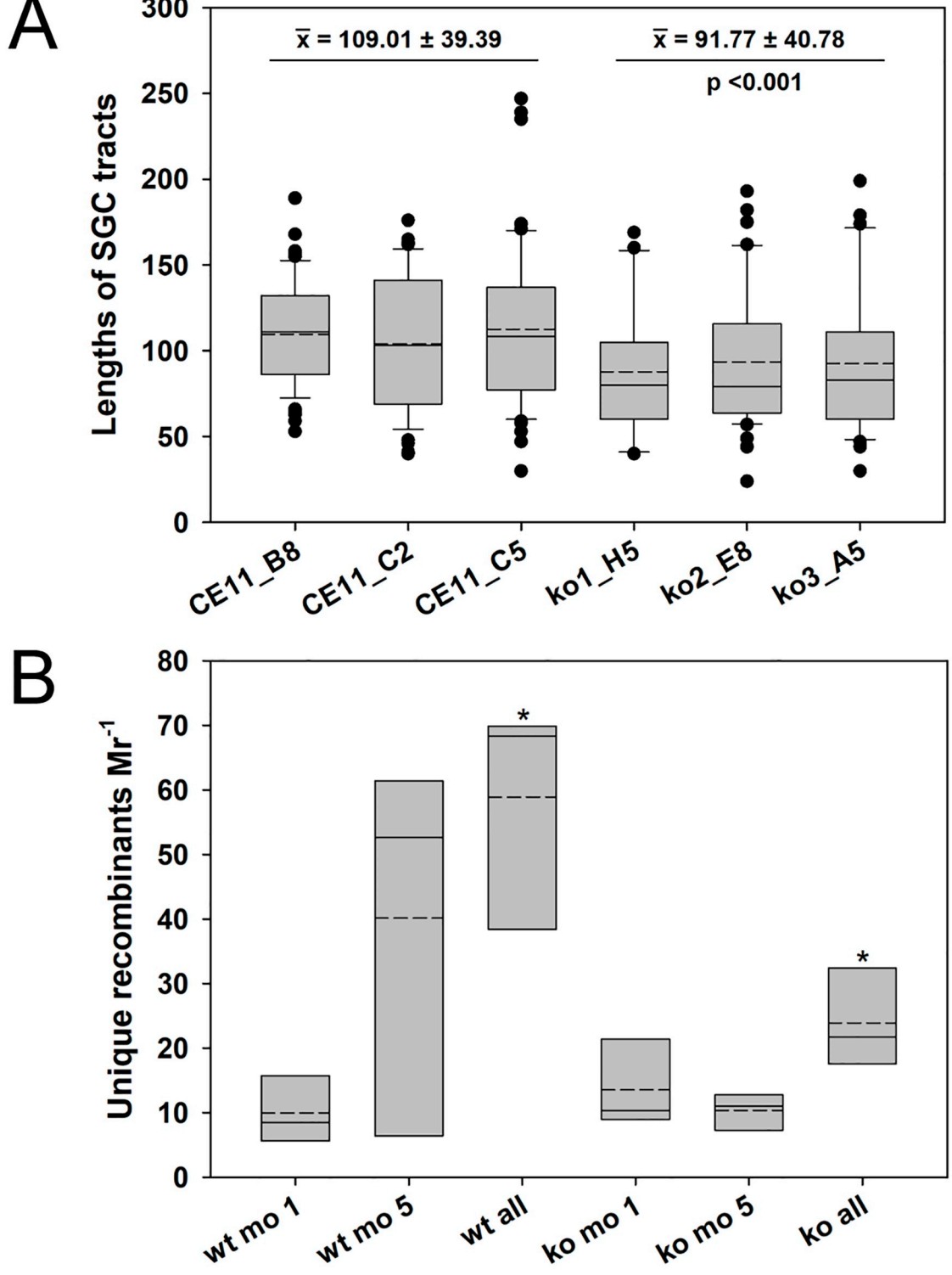

**Fig 5. Lengths and total variation of SGC tracts in wt and ko parasite populations.** These data effectively demonstrate the frequency and processivity of the SGC process. **A**. The lengths of SGC tracts identified in amplicons with full statistical support were compared for each of the six clonal lines under study. Data from 1 mo. and 5 mo. samples were pooled for each sample. A statistically significant difference (p < 0.001) was observed between wild-type and knockout parasites SGC tracts. **B**. The numbers of unique recombinant SGC tracts with full statistical support were plotted, normalized per million merged reads. This value provides a surrogate measure of the total variation presented by a parasite population. Slotted crossbars indicate means, solid crossbars

indicate medians, and the box boundaries represent 25% and 75% confidence intervals. Asterisk: a statistically significant difference of p = 0.035 was observed between wt and knockout parasites when comparing the total populations of unique recombinants (Student's t-test). wt, pooled CE11 wild-type samples; ko, pooled knockout parasite samples.

### Involvement of homologous sequences flanking SGC breakpoints

We wished to assess whether there is anything shared at SGC breakpoint sites, or unique where SGC tracts differed from the active locus of *ves* transcription (LAT). To do this, alignments were made of pairs of sequences identified as having given rise to a unique recombinant, and the recombinant sequence itself. Regions of homology between all three sequences were then identified manually that represented a region in which transition occurred from one parent locus providing the sequence to the other (Fig 6A). No significant differences were observed among wild type or Bb*rad51* knockout genotype parasites in the lengths of homology patches (Fig 6B), and no specific sequences were associated with homology patches. Importantly, for SGC to occur patches of homology between donor and recipient may not be required, as patches of as little as 2 bp were observed. In a few instances no transition region was present, and in still others there was a brief patch in which no match existed between the three sequences at the site of transition, suggesting a sometimes chaotic process (Fig 6A). While this is clearly not an exhaustive analysis of all SGC tracts, the results of this subsampling indicate that there is no apparent difference between wild type and Bb*rad51* knockout parasites with regard to the lengths of the homologous patches possibly involved in recombination.

## Discussion

SGC is a major mechanism of antigenic variation in *B. bovis*, and to date the only one that has been demonstrated critically [12]. This phenomenon, in which short DNA patches are duplicated from a donor to a recipient gene, typically occurs without modification of the donor and at least superficially resembles HR-mediated DNA repair. DNA repair in apicomplexan and other protozoal parasites overall is not well understood [35, 36]. It is even difficult to predict from the parasites' proteomes which repair pathways may be active. Orthologs are recognizable for only a fraction of the proteins known to be important in higher eukaryotes, and many orthologs are simply not present. For example, the proteins RPA, Rad51, Rad52, Rad54, and ATM are considered key participants in DNA repair pathways, including HR and gene conversion. Yet, in *B. bovis* orthologs may be identified only for RPA, Rad51, and Rad54, and many similar examples of "missing" proteins hold [13, 31], suggesting the merging of functions. In this study, we wished to understand the contributions of BbRad51 to SGC because proteins of this family are considered essential to HR and gene conversion in other systems, including other apicomplexans [29, 37]. Our prior identification of BbRad51 as the true Rad51 ortholog was based upon several criteria, including sequence and structural similarities to established Rad51 proteins, a greatly reduced or abrogated ability to achieve HR-dependent integration of exogenous sequences in the absence of BbRad51, and enhanced sensitivity to MMS in knockouts that could be complemented by Bb*rad51* coding sequences [31].

In contrast with our prior work, we present evidence herein that the absence of BbRad51 does not influence the parasite's in vitro survival or extent and rate of general repair of dsDNA breaks engendered by acute exposure to γ-irradiation. The apparent insignificance of BbRad51 to repair and survival of IR-induced DSBs may be attributable to the haploid nature of the asexual stages studied here. A large proportion of IR-caused DSBs would occur in unique regions of the genome, with no intact second copy of the damaged sequence available to support true gene conversion, except briefly during mitosis. Thus, in the absence of available

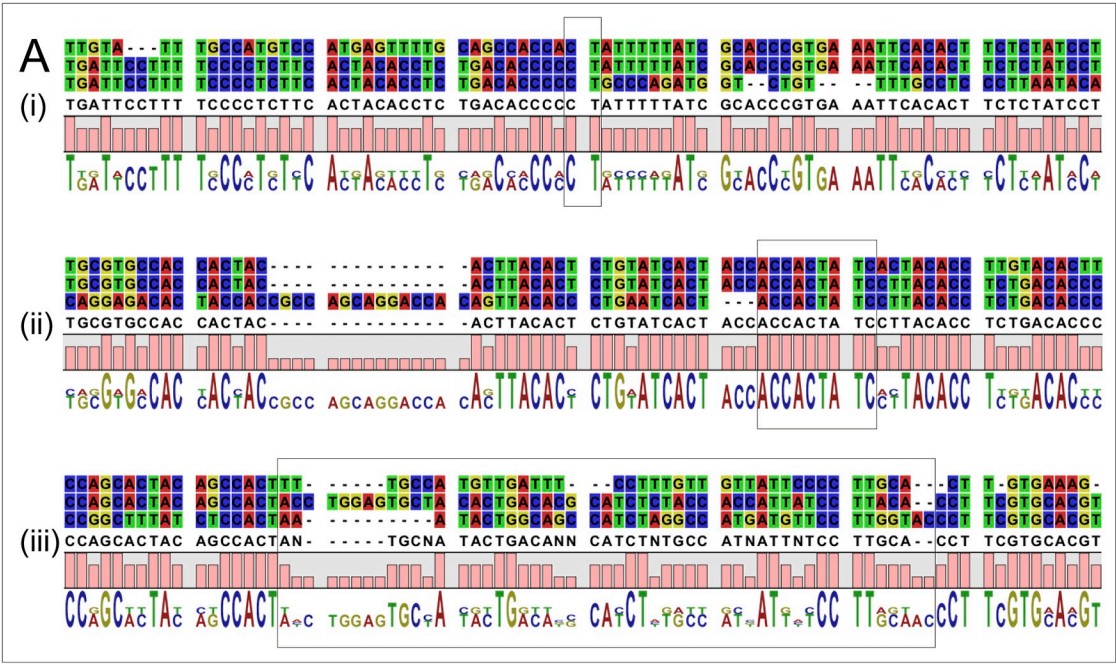

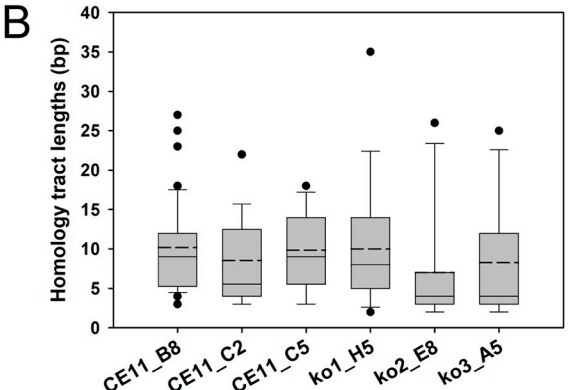

**Fig 6. Junctional sequences observed at recombination sites.** These data reflect the sequence requirements for the SGC process to achieve productive recombination. **A**. Alignment of donor, recipient, and recombinant sequences. In each case, the top sequence is the minor parent (donor), the bottom the major parent (recipient), and the middle sequence is the recombinant sequence. The gray boxes identify the regions within which recombination presumably occurred. Illustrated are patches of (A) 2 bp, (B) 9 bp, and (C) an example wherein no identifiable patch of homology may be found. **B**. Patches of homologous sequence were observed at the breakpoints between the sequences of transcripts representing loci that had served as sequence donors and those of the recipient loci during SGC modifications of actively transcribed *ves1α* genes. These tracts of homology were identified through alignments of donor and recipient sequences as in panel A, and plotted for a minimum of 15 recombinant sequences from each sample. Slotted crossbars indicate means, solid crossbars indicate medians, and the box boundaries represent 25% and 75% confidence intervals. No statistically significant differences were observed between any of the samples (p = 0.133; Kruskal-Wallis one way analysis of variance on ranks).

sequence donors for HR, BbRad51 may be largely superfluous to surviving heavy dosages of IR, yielding similar ionizing radiation survival outcomes in wild-type and knockout parasites. *B. bovis* may depend instead upon error-prone end-joining reactions for survival of such significant damage. Unlike most eukaryotes, including *Toxoplasma* [38], *B. bovis* lacks genes for key players in canonical non-homologous end-joining repair, such as Ku70/80 and DNA ligase 4 [13]. This parasite instead may depend upon a synthesis-dependent microhomology-

mediated end-joining mechanism like that demonstrated in *P. falciparum* [39], but this remains to be determined. In contrast with ionizing radiation, the absence of BbRad51 does render *B. bovis* sensitive to alkylation damage caused by acute exposure to MMS [31]. In diploid organisms recovery from either type of insult is typically compromised by loss of Rad51 [40]. However, in diploids such repair makes frequent use of the second allele for repair through gene conversion. Unlike IR, MMS does not directly cause DSBs, but rather alkylates adenosine and guanidine bases which must be removed and replaced [32]. DSBs still may result when abasic sites or single-strand breaks created during base excision repair of the methylated bases stall or cause the collapse of mitotic replication forks [41]. In diploid cells HR is the major mechanism used in repairing such DSBs during replication, and in at least one mechanism makes use of Rad51 [40, 42]. In TbRad51-intact *T. brucei* (a diploid parasite), the repair of DSBs created within *VSG* bloodstream-stage expression sites resulted in a massive increase in the rate of *VSG* switching through gene conversion, presumably due to the ready availability of related sequences [43]. This also may be accomplished in haploid organisms, if appropriate sequences have already been replicated on the opposite strand, but would limit the timing of gene conversion only to S phase, prior to separation of sister chromatids.

The distribution and structure of *ves* genes might suggest that they have evolved for efficient application of the SGC mechanism. In most microorganisms that undergo antigenic variation the variant multigene families involved typically are arranged in large, subtelomeric clusters [44]. By contrast, *ves* genes are found in numerous small clusters scattered throughout the genome, and often are interspersed with *smorf* genes [10, 13]. Despite extremely high overall variability, *ves* genes possess periodic tracts of highly conserved sequence, and *ves* genes on the same or different chromosomes may provide targets for strand invasion via such conserved tracts [12]. A small-scale chromatin conformation capture (3C) query of sequences in proximity to the transcribed *ves* genes of the LAT suggested close proximity primarily to other *ves* genes [16]. In that organization, the ability to find short patches of homologous sequence in which to initiate strand invasion might occur via those conserved patches. However, when the sequences flanking the breakpoint sites were compared, patches of homologous sequence unrelated to the highly conserved sequence regions were observed at the SGC tract breakpoints between the LAT and donor sequences. These stretches of homologous sequence ranged from 2 bp to 35 bp in length (Fig 6), and in some sequences no identifiable homologous sequences could be identified. These data suggest that, for SGC to proceed only local microhomology is required, and perhaps no homology. While these data do not specifically identify the process responsible they are consistent with SGC not relying upon classical HR mechanisms. The gene family- and subfamily-specific conserved tracts found in most *ves* genes instead likely serve other functions, either in the chromatin or in the VESA1 polypeptides they encode. Interestingly, in *P. falciparum* very similar patterns were reported of recombination between *var2CSA* genes associated with adhesion of infected erythrocytes to chondroitin sulfate in placental malaria [45].

When initiating this study we hypothesized that, as Rad51 proteins are considered essential to HR and SGC is thought to be a form of HR, BbRad51 should be essential to this process and its loss should abrogate SGC. Our results disprove our hypothesis of essentiality, but do not preclude a contribution by BbRad51 to the process. The large decrease in number of SGC tracts Mr$^{-1}$ may indicate that BbRad51 participates as an integral component in one of multiple pathways resulting in SGC. However, given the similarity in sequences found in recombination junctions, as reflected by the SGC tract homology patches in wt and knockout parasites (Fig 6), it seems more likely that BbRad51 might be contributing indirectly, perhaps through HR rescue of some proportion of the population following DSBs. The analysis of sequences surrounding SGC breakpoints supports error-free recombination via homologous sequences

in most, but not all, instances. However, unlike Rad51-mediated HR where tracts of homologous sequence at least 8 bp long are needed for synapse formation [46, 47], only local sequence microhomology of at least 2 bp, or perhaps no homology, appears to be needed (Fig 6). It is possible that these junctional traits are an artifactual result arising from misidentification of recombining sequences during analysis. Alternatively, this may be the actual biological result, due to mismatch repair or deletion subsequent to sequence acquisition or replication. To definitively resolve this uncertainty would require finding these identical rearrangements in genomic DNA from the same parasite population, a tall challenge considering the small proportion of the population likely to carry a particular recombination. Whereas these results reflect the mechanism(s) responsible, they do not identify the pathway. The lack of significant differences in the homology patches flanking SGC recombinations is consistent with a lack of direct BbRad51 involvement, and with the loss of the ability to incorporate exogenous plasmid sequences into the genome via HR in the absence of BbRad51 [31]. The reason for the apparent shortening of SGC tracts observed in knockouts is not clear. One possibility is that it may reflect the loss of some recombinants where longer unique tracts were involved, which might require BbRad51 for successful disengagement of the invading and donor strands, repair of breaks created during the disengagement process, or stabilization of a longer crossover region during exchange. So, while BbRad51 is clearly not essential to the SGC process *per se*, it may enhance its efficiency through such secondary effects. A more global potential secondary effect that may be dependent on BbRad51 but cannot be observed using our strategy is effect on overall genome structure. For example, the recombination cascades observed during recombination of *var* genes in *P. falciparum* [48].

The overall reduction in isTS observed among Bb*rad51* knockouts, based upon the numbers of alternative loci to which transcripts mapped in each line, is particularly intriguing. At the 5-month time point, in the knockout lines transcripts appear to have arisen from a mean of only 23.3 *ves* loci, whereas in wild-type parasites a mean of 33.3 loci had been activated within the population. Although statistically significant (p = 0.014), the biological significance of this difference is not as clear. The reason is because these data are derived from *ves1α* genes only. Also, the maximal number of *ves* loci that are competent to be activated is not known and the ability to modify a single locus is extensive. About half of the *ves* family is organized in divergent (head-to-head) pairs (approximately 33 pairs) of *ves1α*/*ves1β* or *ves1α*/*ves1α* genes that flank quasi-palindromic, bidirectional promoter regions [12, 16]. The remainder are present as individual *ves* genes with potentially unidirectional promoters. Among the *ves* loci putatively activated in this study, the ratio of loci with divergent/ unidirectional promoters ranged from 1.07–1.60 (mean 1.28 ± 0.24) in CE11, and 1.25–2.00 (mean 1.61 ± 0.29) in knockouts. Although wild type parasites, on average, appeared to activate a higher proportion of unidirectional promoters than did knockouts, this did not reach statistical significance (p = 0.059; S1 Table). The functionality of several bidirectional *ves* promoters has been demonstrated experimentally [16], but function has not yet been similarly tested for promoters preceding individual *ves* genes. This result clearly indicates that a significant proportion of *ves* loci have the potential to be activated, including individual *ves* genes, consistent with a study on the transcriptomes of pairs of virulent *B. bovis* lines and attenuated lines derived from them. In that study, virulent lines transcribed from a significantly wider variety of *ves* loci that included both divergent and non-divergent loci [49]. Interestingly, among attenuated parasites *ves* transcription was upregulated only from loci that are not divergently-oriented. The difference in *ves* transcriptional behavior observed here and in the attenuation study suggests that in vivo attenuation of *B. bovis* is unlikely to be related to BbRad51 expression or function. This conclusion is supported by the unperturbed Bb*rad51* transcription observed in attenuated parasites [49].

The basis for a connection between the SGC and isTS mechanisms is not clear. Indirect evidence allows us to propose at least three feasible explanations, each with varying levels of support. (i) First, of the various *ves* loci represented among the transcripts, a subset of the sequences may reflect the complete replacement of the observed region within the original locus of active *ves* transcription (LAT) by much longer conversion tracts, rather than by isTS. In this case, such a long replacement sequence would cause the read to map artifactually to the donor locus rather than to the locus from which the full *ves* gene was actually being transcribed. Given the inability to integrate exogenous DNAs into the genome of Bb*rad51* knockout parasites [31] it is anticipated that this would be a rare event in knockouts. From our data, we cannot rule out this possibility. (ii) The sites failing to activate in Bb*rad51* knockouts may be unusually sensitive to recombination, with most such events leading to lethality. However, comparison of all the putatively activated *ves* loci in wild-type and knockout parasites reveals that there is essentially complete overlap in the *ves* gene clusters that can be activated (S2 Fig), directly arguing against this explanation. Rather, the knockouts appear to achieve comparable inter-locus switches, but with a lower frequency than wild-type. (iii) A mechanistically distinct possibility is that Bb*rad51* knockouts have a diminished capacity to activate *ves* loci epigenetically. As a part of DNA repair, chromatin first must be remodeled to make it accessible to the repair machinery. This occurs in part by local chromatin decondensation through histone acetylation [50–52]. Rad51 has been proposed to assist in the assembly of the histone acetylation machinery during repair of dsDNA breaks and stalled replication forks [53]. In the presence of BbRad51, double-stranded breaks in silenced *ves* loci may be successfully acetylated and repaired, but may not always be remodeled again for silencing. With transcription of a single *ves* locus being the default state, a choice would have to be made between the existing LAT (the single active locus of *ves* transcription [15]) and the newly repaired/acetylated *ves* locus. Silencing of the existing LAT would lead to isTS and establishment of a new LAT. Thus, BbRad51 may be epigenetically effecting isTS of *ves* genes as an unintended side-effect of the DNA repair process. If true, then in Bb*rad51* knockouts the absence of BbRad51 would be anticipated to result in frequent failure to assemble the full repertoire of repair machinery, including epigenetic modifiers. Accordingly, isTS would be a less common event. While explanation (i) is consistent with well-established Rad51 protein functions, possibility (iii) is neither implausible nor inconsistent with less well-characterized functions, and even could provide a potential mechanism for the stochastic switch events of isTS. If substantiated experimentally, this could provide a direct link between DNA repair and antigenic variation via isTS as well as recombination. Our currently available evidence does not distinguish these two possibilities, and this question warrants further investigation. It is noteworthy that these results are similar to observations in *T. brucei*, which possesses five Rad51 protein family genes. Knockouts of Tb*Rad51* and Tb*Rad51-3* each were shown to result in diminution of *VSG* gene switch events not dependent upon DNA recombination, as well as the expected HR-associated reductions [28, 54]. This observation suggests that such proposed interactions of Rad51 proteins with the epigenetic machinery may be widespread phylogenetically, but not yet widely appreciated.

The limitations of our study include the lack of genomic data for our culture populations. High coverage whole genome sequencing, and good quality assemblies might improve the assessment of SGC and recombination. Still, our results demonstrate that BbRad51 is not necessary for survival of asexual *B. bovis* in vitro or for overall genome stability in the absence of environmental insult. Moreover, this protein is dispensable to SGC-based antigenic variation in *B. bovis*, although it influences the rates of SGC antigenic variation and isTS. It is not clear whether its absence would be similarly benign during in vivo infection, where there is strong selection by host immune responses. Evolutionary retention of BbRad51 and its involvement in recovery from alkylation damage indicates clearly that it plays some role(s) in parasite DNA

repair, including in asexual developmental stages. The clear implication of this work is that some component(s) besides BbRad51 provides for the recombination observed in SGC. Whether this is from a more distantly related member of the RecA/RadA/Rad51 superfamily proteins encoded by the *B. bovis* genome, or another enzyme class altogether, awaits experimental evidence.

## Materials and methods

### Parasite culture, transfection, and selection

This project used the clonal *B. bovis* CE11 parasite line as starting material [6]. In vitro parasite cultures were maintained as microaerophilous stationary phase cultures under 90% $N_2$/ 5% $O_2$/ 5% $CO_2$ (v/v), essentially as described [5, 55]. Cloning of parasites was conducted by two sequential rounds of limiting dilution cloning as described previously [7]. Parasites were transfected with DNAs purified from *E. coli* DH5α, using EndoFree Plasmid Maxi kits (Qiagen; Valencia, CA). Both parasitized erythrocytes and DNAs were suspended in cytomix (120 mM KCl, 0.15 mM $CaCl_2$, 10 mM $K_2HPO_4$/$KH_2PO_4$ pH 7.6, 25 mM HEPES pH 7.6, 2 mM EGTA, 5 mM $MgCl_2$, pH 7.6) [56] prior to electroporation. Electroporation was performed in 2 mm-gap cuvettes, using 5 pmol linearized DNA at 1.25 kV, 25 μF, and 200 Ω, as described by Wang et al. [16], with plating at 1.25% packed cell volume. After 24h recovery in culture, selection was initiated by addition of blasticidin-s hydrochloride (TOKU-E; Bellingham, WA) to a final concentration of 15 μg ml$^{-1}$ (32.7 μM) [57]. Every three days medium was removed, and replaced with a 2.5% packed cell volume of uninfected erythrocytes in medium plus blasticidin-s. Once viable parasites emerged, usually after approximately two weeks, they were maintained under drug selection and were immediately cloned.

### Validation of Bbrad51 knock-out

Validation of Bb*rad51* gene knock-out was performed by diagnostic PCR, Southern blotting, RT-PCR, and sequencing of the Bb*rad51* locus. These data are presented in [31].

## Phenotypic analyses

### Parasite growth assays

Parasite growth was assayed by counting Giemsa-stained smears, with samples collected at 0, 24, and 48h growth (approximately 0, 3 and 6 cell cycles [58]). Alternatively, in some experiments a DNA-based SYBR Green I method was performed, essentially as described [59, 60], on parasites grown in bovine erythrocytes depleted of leukocytes [61]. For experiments involving γ-irradiation, parasites were exposed to a calibrated [Cs$^{137}$] source (Gammacell GC-10 gamma irradiator), on ice. Control cells were maintained on ice for the duration of the treatment time. Samples were immediately diluted into fresh medium containing 10% packed cell volume uninfected erythrocytes, and placed into culture. For experiments involving MMS exposure, parasites were exposed to MMS (diluted in complete medium) for 90 minutes, followed by washout as described [31].

### Chromosome reassembly

Parasites grown in leukocyte-depleted erythrocytes as described above. Cultures, at 2.5% parasitized erythrocytes, were given 100 Gy exposure to [$^{137}$Cs] on ice to fragment chromosomes. Irradiated cells were placed back into culture to recover for designated times, then were processed for pulsed-field gel analysis [62]. Plugs were embedded into 1% SeaKem Gold agarose in 0.5x TBE buffer, and electrophoresed for 23.5 hours at 180V, with a 50–165 second ramped

switch time [63]. Gels were stained with SYBR Gold Nucleic Acid Stain (Invitrogen) for DNA visualization, and photographed. Integrated pixel intensities were plotted for each chromosome and the "smear" of DNA below chromosome 1 using ImageJ v. 1.52 "Gels" and "Measure" algorithms. Corresponding blank gel regions were used for background correction.

## SGC-mediated antigenic variation assay

A schematic representation of this experiment is provided in Fig 3 for clarity. This experiment was performed with three biological replicates per genotype, comprised of one clone each from three independent Bbrad51 knockout lines (CE11Δrad51[ko1] H5, CE11Δrad51[ko2] E8, and CE11Δrad51[ko3] A5; referred to as ko1[H5], ko2[E8], and ko3[A5]), and three subclones of wild type CE11 line parasites (CE11 B8, CE11 C2, and CE11 C5). Bb*rad51* knockout and wild type parasites were cloned by limiting dilution [7]. RNAs were isolated one and five months after parasite cloning, using Ribozol (Amresco). RNAs were treated two times with TurboDNase (Ambion) supplemented with 1 mM $MnCl_2$ [64], followed by inactivation with DNase Inactivation Reagent (Ambion). M-MuLV reverse transcriptase (New England Biolabs) and oligo-d (T) primers were used to make cDNAs. A hypervariable segment containing most of the CKRD domain of *ves*1α transcripts was amplified by RT-PCR, using "universal" primers vesUniF2 (TGGCACAGGTACTCAGTG) and PD1R (TACAANAACACTTGCAGCA) as described [15].

## Sequencing and recombination analysis

Four independent amplifications of each cDNA were pooled in stoichiometrically equal amounts to maximize detection of rare variants and minimize single-sample PCR artifacts during sequencing. Paired-end amplicon libraries were generated with NEBNext reagents (New England Biolabs) by the University of Florida NextGen Sequencing Core Laboratory, incorporating Illumina TruSeq index sequences. Libraries, spiked with 8% PhiX genomic library as internal control, were sequenced on the Illumina MiSeq platform, using the Illumina Pipeline 1.8. Fastq reads were analyzed using Qiime2 pipeline [65]. Quality and adapter trimming were performed using CutAdapt [66, 67]. Further de-noising and amplicon sequence variant (ASV)-calling were performed using DADA2 [68], truncating the reads at 230 nt and allowing a maximum of 5 expected errors per read. In order to remove potential contaminating sequences, ASVs were aligned with the bovine genome, using BLAST [69]. In order to identify recombinant sequences, ASVs were first aligned using CLC Main Workbench, version 6.9.2. Recombination analyses were then performed on the alignments with RDP, GENE-CONV, Bootscan, Maxchi, Chimaera, SiSscan, and 3Seq, as implemented in RDP4 [70]. Only recombinant events identified by a minimum of 4 out of seven tests (at p ≤0.05) were considered statistically supported and included in downstream analyses. Note that no genome is currently available for the CE11 line. Therefore, the C9.1 line genome was used, as the C9.1 and CE11 lines are sibling clonal lines derived from the MO7 clonal line [6, 7, 12]. Recombinant results, including extraction of SGC tracts, were summarized using custom scripts written with R version 3.6.3 [71] through the RStudio shell. Mean lengths of SGC tracts were compared by one-way ANOVA, whereas tract length distributions were compared by the Mann-Whitney Rank Sum test, without expectation of a normal distribution of variance, using SigmaPlot version 11.0 (Systat Software, Inc.; San Jose, CA).

## Supporting information

**S1 Fig. Titration of γ-irradiation dosage on *B. bovis* CE11 (wild type).** Parasites were exposed to 0–1000 Gy irradiation from a calibrated $Cs^{137}$ source, then placed back into culture

and the percent parasitized erythrocytes determined from Giemsa-stained smears made at 0, 24, and 48h growth. For experiments requiring survival of a proportion of the parasites the 100 Gy dosage was chosen.
(TIF)

**S2 Fig. Locations of *ves* loci from which transcripts and SGC tracts arose.** The genomic locations of *ves* loci to which amplicons mapped (plotting the full 8 Mbp genome as a single linear element), and from which they were presumed to be transcribed, is plotted relative to the numbers of parasite lines transcribing from that locus. No *ves* gene clusters within the genome were transcribed by CE11 parasites that could not also be transcribed by knockout parasites, and vice versa. For this plot, data were pooled from both time-points of all three wt or all three knockout lines (i.e., maximally 6 possible per *ves* locus).
(TIF)

**S3 Fig. Statistically supported recombinants *vs*. total unique amplicons.** A regression plot was made of the frequencies of merged amplicons with full statistical support for identification as true recombinants against the total numbers of unique amplicons observed for each population, with each type of value normalized per million reads ($R^2 = 0.9228$).
(TIF)

**S4 Fig. Cumulative distribution of unique SGC tract lengths in wild-type and knockout parasites.** The length distribution of unique SGC tracts was plotted as the cumulative proportion of total SGC tracts against tract lengths. Samples were pooled for all six samples for wild-type and knockout parasites. These pooled data are from the same CE11 wild-type and knockout lines shown in Fig 4B. The vertical index lines indicates the median values of each population. Differences among the distributions in each sample type were determined by the Mann-Whitney Rank Sum test ($p = <0.001$).
(TIF)

**S5 Fig. Unique SGC tracts per active *ves* locus.** The numbers of unique SGC tracts with full statistical support per transcriptionally active *ves* locus is shown for *B*. *bovis* CE11 wild-type and knockout populations at 1 and 5 months growth post-cloning. The data were pooled from all three lines of each population type. Slotted crossbars indicate means, solid crossbars indicate medians, and the box boundaries represent 25% and 75% confidence intervals. Values did not vary significantly among samples ($p = 0.986$, based upon one-way ANOVA).
(TIF)

**S1 Table. Organizational nature of the *ves* loci to which reads mapped.** Loci to which amplicons mapped are listed for all 12 samples (i.e., 1-month and 5-month RNAs from all three wild-type and knockout lines), along with mean lengths and the nature of each locus. Both divergent, bidirectional loci and unidirectional loci were active in transcription at some level in all samples.
(DOCX)

**S1 File. This .zip file contains individual files of SGC tracts identified for all six clonal lines, in .csv format.**
(XLSX)

## Acknowledgments

The authors thank Allison Vansickle for assistance with animal care and handling, and Kevin Brown, Linda Bloom, Salvador Castaneda Barba, Eva Top, and Olivia Kosterlitz for helpful comments.

## Author Contributions

**Conceptualization:** Erin A. Mack, David R. Allred.

**Data curation:** David R. Allred.

**Formal analysis:** Erin A. Mack, Massimiliano S. Tagliamonte, David R. Allred.

**Funding acquisition:** David R. Allred.

**Investigation:** Erin A. Mack, Yu-Ping Xiao, Samantha Quesada, David R. Allred.

**Methodology:** Erin A. Mack, Massimiliano S. Tagliamonte, Yu-Ping Xiao, David R. Allred.

**Project administration:** David R. Allred.

**Resources:** David R. Allred.

**Supervision:** David R. Allred.

**Writing – original draft:** Erin A. Mack, David R. Allred.

**Writing – review & editing:** Erin A. Mack, Massimiliano S. Tagliamonte, David R. Allred.

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
