## [Decision Letter · Decision Letter 0]

28 Jul 2020

Dear Professor Allred,

Thank you very much for submitting your manuscript "Babesia bovis Rad51 ortholog influences switching of ves genes but is not essential for segmental gene conversion in antigenic variation" for consideration at PLOS Pathogens. As with all papers reviewed by the journal, your manuscript was reviewed by members of the editorial board and by several independent reviewers. The reviewers appreciated the attention to an important topic. Based on the reviews, we are likely to accept this manuscript for publication, providing that you modify the manuscript according to the review recommendations.

You will see that the reviewers offer primarily minor suggestions for you to address. The one exception is the major issue noted by reviewer 2 regarding the experiments with the Rad51 fusion protein and the use of the glmS system to knockdown expression. The reviewer correctly notes that the genetic modifications might have affected expression levels or enzymatic activity, and these possibilities have not been controlled for. In the Plasmodium field, the use of glmS has recently been found to sometimes cause a significant knockdown even in the absence of glucosamine, thus the concerns are warranted. However, the reviewer also notes that this set of experiments are not imperative for the primary conclusions of the paper, and thus could be omitted if the proper controls cannot be added. 

Sincerely,

Kirk Deitsch

Section Editor

PLOS Pathogens

Kasturi Haldar

Editor-in-Chief

PLOS Pathogens

orcid.org/0000-0001-5065-158X

Michael Malim

Editor-in-Chief

PLOS Pathogens

orcid.org/0000-0002-7699-2064

Reviewer Comments (if any, and for reference):

Reviewer's Responses to Questions

**Part I - Summary**

Reviewer #1: This paper describes the role of Rad51 in antigenic variation in the haploid eukaryotic parasite Babesia bovis that depends on a mixture of active site switching (isTS) and segmental gene conversion to great novel mosaic antigen genes (ves genes). This same group previously published data from their Rad51 knockout parasites describing a predictable phenotype in the Rad51 knockout parasite line of mutagen sensitivity and defects in HR that necessitated the development of an artificial chromosome in order to establish a complemented Rad 51 parasite line.

In this work, the group focuses on the potential role of Rad51 in antigenic variation. The results are somewhat unexpected as segmental gene conversion continued to occur as a mechanism of creating novel ves genes and in contrast to their earlier data with the mutagen MMS, DNA damage induced by irradiation was repaired equally in the knock out line. The most pronounced phenotype was the ability of the parasite to switch active ves sites implicating a potentially unique role for Rad51 in gene expression and silencing and the existence of an alternative pathway driving the diversification of ves genes. This paper adds to a body of work demonstrating alternative roles in eukaryotic pathogens for well conserved DNA repair proteins such as the recently described role in replication demonstrated in L. major (Damasceno et al 2020).

Most other parasitology groups have used Western blots and RT-PCR approaches to examine parasite responses to DNA damage and in this data has been somewhat muddy. I appreciate that this group took the time to pursue a different approach with the potential to separate the confounding aspect of parasite death in determining if protein levels were upregulated in response to DNA damage. I also appreciate that they showed data in Figure 2 in two different formats to better assess the impact of irradiation on parasite growth.

The fact they did not find significantly different recombination in the knockout line is quite interesting given the known role of gene conversion in this process. They refer to the process as SGC even as they demonstrate that there is minimal to no homology at recombination sites and even the formation of completely novel sequence. This appears closer to VDJ recombination and somatic hypermutation than to SGC and, as their data shows, is not Rad51 dependent. Is there a potential for a mixture of DNA recombination pathways contributing to the ves gene diversification and should be referred to differently?

They use data from independent clones at different points in time and a conservative approach to determine a true recombinant sequences from their data show an appropriate amount of rigor to their results. They mention that there was no difference in recombination at >150 bp tract lengths- given their approach was to amplify a segment I am slightly concerned there may be some potential bias in their approach in that they may have missed larger recombination events?

Reviewer #2: This is an interesting, valuable study that represents, to the best of my knowledge, the first attempt to genetically test for the mechanisms involved in antigenic variation in the apicomplexan parasite Babesia bovis. Antigen variation typically occurs in other pathogens by changing the single surface antigen gene amongst a family that is transcribed, or by recombination of silent antigen genes into dedicated sites of expression. B. bovis is interesting because both strategies are employed, making it quite different from Plasmodium falciparum. In fact, simultaneous transcription- and recombination-mediated antigenic variation is also found in Trypanosoma brucei and the results presented suggest intriguing parallels, in that loss of the key factor of homologous recombination, Rad51, impairs (but does ablate) both reactions. I have only a few minor suggestions to make in what is overall a comprehensive manuscript.

**Part II – Major Issues: Key Experiments Required for Acceptance**

Reviewer #1: none

Reviewer #2: 1. I’m afraid that I am not certain what we have learned from the first set of experiments described, in which the authors translationally fuse Rad51 to Nanoluciferase and HA and test for expression before and after damage, and with and without of transcript via glmS. The authors say that they wish to ask if MMS damage results in increased Rad51 expression but do not explain why they expect to see this. More importantly, they do not show that the genetic modifications used to monitor expression do not result in impaired gene expression or loss of protein function, both of which would undermine the findings. They should compare levels of Rad51 mRNA, by RT-qPCR, in wild type and modified cells, with and without exposure to MMS. In addition, they should compare MMS sensitivity of the modified cells (expressing the rad51-HA fusion) to wild type cells and rad51 mutants to ensure the modified rad51 protein is functional.

**Part III – Minor Issues: Editorial and Data Presentation Modifications**

Reviewer #1: I would think that Figues 5, 6 and 7 could be combined though it would create a long figure legend. Either way, if possible Supplemental Fig 7 should be part of the main body of the paper.

Page 10 line 279- can you clarify why not feasible? Perhaps rephrase

Page 12 line 336 – typo

page 15 line 436- did you mean supp fig7?

Page 15 line 437- the authors humbly acknowledge the limitations of their study. It would be nice to see a follow up where they state how they would/will address in the future perhaps with WGS.

Reviewer #2: 2. To my mind the title of the section ‘Bbrad51 knockout did not prevent in situ transcriptional switching’ is misleading. While what is said is correct, the title fails to document the fact that mutation of rad51 leads to lower levels of ves genes transcribed over time. The abstract much more clearly documents this important finding.

3. I would suggest that a diagram summarising how the authors have assayed SGC is needed, as this is very hard to determine from results and methods, and is important for the reader to understand the data.

4. Again, I think that the title of the section ‘Bbrad51 knockout failed to prevent segmental gene conversion’ is misleading (or describes only part of the data). The authors clearly document that the rad51 mutants display a reduced length of gene conversion tract, and appear to show lower levels of recombination over time. Both these data indicate that loss of the factor does impair antigenic variation, which is not captured by the section title.

5. The concern in point 4 filters into the discussion: e.g. line 429 ‘When initiating this study we hypothesized that, as Rad51 proteins are considered essential to HR and SGC is thought to be a form of HR, BbRad51 should be essential to this process and its loss should abrogate SGC. Our results disprove that hypothesis’. This description is, to my mind, rather too ‘all or nothing’, and does not adequately capture the complexity of the data (unlike the well worded abstract). Indeed, the authors appear to be missing a rather obvious conclusion from their data when they discuss (lines 429-449) the effects seen in the rad51 mutant and state, ‘Regardless of any secondary effects, BbRad51 is clearly not essential to the SGC process’. Surely the fact that both gene conversion track length and the extent of ves recombination is reduced in the mutants suggest that rad51 does play a role in SGC but that its loss allows an unknown, less efficient reaction to direct ves recombination (e.g. with different homology requirements or catalysis mechanism)?

Very minor points:

6. In line 382 where in ‘this study’ is it documented that Ku and Ligase 4 are absent in Babesia?

7. The change in in situ transcription in the rad51 mutants described in this work is intriguing and well discussed. Has a similar effect been described in any other pathogen that relies on transcription for antigenic variation, such as Plasmodium or Trypanosoma? In other words, is this finding likely to have implications beyond Babesia?

PLOS authors have the option to publish the peer review history of their article (what does this mean?). If published, this will include your full peer review and any attached files.

Reviewer #1: No

Reviewer #2: No
---

## [Editor Report · Decision Letter 1]

13 Aug 2020

Dear Professor Allred,

We are pleased to inform you that your manuscript 'Babesia bovis Rad51 ortholog influences switching of ves genes but is not essential for segmental gene conversion in antigenic variation' has been provisionally accepted for publication in PLOS Pathogens.

Best regards,

Kirk W. Deitsch

Section Editor

PLOS Pathogens

Kirk Deitsch

Section Editor

PLOS Pathogens

Kasturi Haldar

Editor-in-Chief

PLOS Pathogens

orcid.org/0000-0001-5065-158X

Michael Malim

Editor-in-Chief

PLOS Pathogens

orcid.org/0000-0002-7699-2064
---

## [Editor Report · Acceptance letter]

27 Aug 2020

Dear Professor Allred,

We are delighted to inform you that your manuscript, "Babesia bovis Rad51 ortholog influences switching of ves genes but is not essential for segmental gene conversion in antigenic variation," has been formally accepted for publication in PLOS Pathogens.

Best regards,

Kasturi Haldar

Editor-in-Chief

PLOS Pathogens

orcid.org/0000-0001-5065-158X

Michael Malim

Editor-in-Chief

PLOS Pathogens

orcid.org/0000-0002-7699-2064